# Transcriptome and metabolomics analysis of adaptive mechanism of *Chinese mitten crab* (*Eriocheir sinensis*) to aflatoxin B1

Hongsheng Yang[1,2‡], Meifang Shen[1], Qiuyun Zhang[2], Yifeng Li[3], Xiuhui Tan[2], Xuguang Li[2], Huimin Chen[4], Lei Wu[5], Shaofang He[5], Xiaohua Zhu[1] *

**1** Fishery Analysis and Testing Center of Jiangsu Province, Nanjing, Jiangsu, China, **2** Freshwater Fisheries Research Institute of Jiangsu Province, Nanjing, Jiangsu, China, **3** College of Aquatic and Life Sciences, Shanghai Ocean University, Shanghai, China, **4** SCIEX Analytical Instrument Trading Co., Shanghai, China, **5** Yitian Technologies Corporation, Nanjing, Jiangsu, China

‡ HY first author to this work.
* xhz824@sina.com

**Data Availability Statement:** All relevant data are within the paper and its Supporting Information files.

## Abstract

Aflatoxin B1 (AFB1), with the strong toxicity and carcinogenicity, has been reported to great toxicity to the liver and other organs of animals. It cause huge economic losses to breeding industry, including the aquaculture industry. *Chinese mitten crabs* (*Eriocheir sinensis*), as one of important species of freshwater aquaculture in China, are deeply disturbed by it. However, the molecular and metabolic mechanisms of hepatopancreas and ovary in crabs underlying coping ability are still unclear. Hence, we conducted targeted injection experiment with or without AFB1, and comprehensively analyzed transcriptome and metabolomics of hepatopancreas and ovary. As a result, 210 and 250 DEGs were identified in the L-C vs. L-30 m and L-C vs. L-60 m comparison, among which 14 common DEGs were related to six major functional categories, including antibacterial and detoxification, ATP energy reaction, redox reaction, nerve reaction, liver injury repair and immune reaction. A total of 228 and 401 DAMs in the ML-C vs. ML-30 m and ML-C vs. ML-60 m comparison both enriched 12 pathways, with clear functions of cutin, suberine and wax biosynthesis, tyrosine metabolism, purine metabolism, nucleotide metabolism, glycine, serine and threonine metabolism, ABC transporters and tryptophan metabolism. Integrated analysis of metabolomics and transcriptome in hepatopancreas discovered three Co-enriched pathways, including steroid biosynthesis, glycine, serine and threonine metabolism, and sphingolipid metabolism. In summary, the expression levels and functions of related genes and metabolites reveal the regulatory mechanism of *Chinese mitten crab* (*Eriocheir sinensis*) adaptability to the Aflatoxin B1, and the findings contribute to a new perspective for understanding Aflatoxin B1 and provide some ideas for dealing with it.

## Introduction

Aflatoxin is a secondary metabolite produced by Aspergillus flavus and Aspergillus parasiticus, and is a highly toxic natural pollutant widely present in feed in humid and hot areas [1, 2].

**Funding:** The author(s) received no specific funding for this work.

**Competing interests:** The authors have declared that no competing interests exist.

Among the 12 aflatoxin derivatives detected, the most toxic and carcinogenic aflatoxin B1 (AFB1) is the most common, with significant toxicity to the liver and other organs of animals [3].

In recent years, aflatoxin pollution has caused huge economic losses to breeding industry, including aquaculture industry [4]. According to data from the Food and Agriculture Organization (FAO) of the United Nations, approximately 25% of crops worldwide are contaminated with varying degrees of aflatoxin every year, and about 2% of crops lose their feed value due to severe pollution [5, 6].

The effects of AFB1 on aquatic animals have been widely reported in many research articles both domestically and internationally, such as *Oreochromis niloticus* [7, 8], *Oncorhynchus mykiss* [9], *Litopenaeus vannamei* [10], *Lctalurus punctatus* [11], *Labeo rohita* [12] and other species. According to reports, due to feeding *Oreochromis niloticus* with AFB1 for 10 weeks, the growth rate slows down and the mortality rate gradually increases [13]. The reproductive capacity of *Carassius auratus gibelio* fed with AFB1 is severely affected [14]. Current research shows that AFB1 can cause abnormal behavior, slow growth, immunosuppression, antioxidant damage and tissue lesions of aquatic animals [15–17]. If AFB1 accumulates in the body tissues of aquatic animals and spreads further in the ecosystem along with the food chain, it may threaten human health [18]. Therefore, the agricultural industry standard of the People's Republic of China (NY5072-2002) stipulates that the safety limit of AFB1 in fishery composite feed is 10μg/kg [19].

*Chinese mitten crab* (*Eriocheir sinensis*) is one of the important species in freshwater aquaculture in China, which has produced good economic benefits [20]. In recent years, with the continuous expansion of aquaculture scale and the continuous improvement of intensification, various diseases have emerged in Chinese mitten crab, seriously hampering its healthy development [21]. As mentioned above, we are aware that AFB1 has significant toxicity to animal livers [3]. We analyzed the obtained data in this experiments and explained some molecular response mechanism in the results. The 400 μl aflatoxin B1 of standard solution was injected into the tested crabs, and the experimental data of hepatopancreas and ovary within one hour after treatment were obtained through transcriptome sequencing and metabonomics methods. The results of this experiment provide data support for our study on the stress response of aflatoxin B1 to crabs, and provide ideas for solving the practical production problems of aflatoxin B1 on crabs in the future.

## Materials and methods

### Experimental reagent

The aflatoxin B1 standard, with a purity of 99.8%, was provided by Israel FERMENTEK Company. The Dimethyl sulfoxide (chromatographically pure) was provided by ROE Company of the United States. The standard AFB1 working solution, with a concentration of 60 mg/L, was prepared by weighing 6 mg of aflatoxin B1 standard and dissolving it with 100 ml of dimethyl sulfoxide. The blank working solution was prepared using a single dimethyl sulfoxide reagent.

### Experimental design and sample collection

More than 18 female experimental crabs were raised at the Pukou Breeding Base in Nanjing, Jiangsu Province, China. Before the experiment, 18 female crabs with an initial average weight of (125±12g) were randomly divided into three groups. Two groups of female experimental crabs were injected with disposable needles for 400 μl AFB1 standard solution with a concentration of 60 mg/L, and the third group was injected with 400 μl blank working solution. The hepatopancreas and ovaries of experimental crabs injected with AFB1 in first group were

taken out after 30 minutes, and those in the second group were taken out after 60 minutes. The hepatopancreas and ovaries of experimental crabs injected with blank working solution were taken out as control after 30 minutes. The samples were frozen with liquid nitrogen and quickly stored in an ultra-low temperature refrigerator (-80˚C). Six samples in each group were used for metabolomics, and three samples in each group were randomly selected for transcriptome sequencing.

## RNA extraction and transcriptome sequencing

Total RNA was extracted from frozen samples using Trizol reagent method (DP762-T1C). RNA purity and integrity were assessed by a NanoPhotometer® spectrophotometer (IMPLEN, CA, USA) and a RNA Nano 6000 Assay Kit on the Agilent Bioanalyzer 2100 system (Agilent Technologies, CA, USA). RNA contamination and degradation were assessed by 1.5% agarose gel electrophoresis.

A total of 1 μg of RNA per sample was used as the input material for library preparation. The mRNA was purified from the total RNA using poly-T oligo-attached magnetic beads. Sequencing libraries from the above three groups (one control and two treatments) were constructed with the purified mRNA using the NEBNext® UltraTM RNA Library Prep Kit for Illumina® (NEB, USA). According to the manufacturer's recommendations, The cDNA library preparations for next-generation sequencing were generated from each sample. Each cDNA library was purified with AMPure XP system (Beckman Coulter, Beverly, MA, USA) and its quality was assessed by using Agilent RNA 6000 Nano Kit on the Agilent Bioanalyzer 2100 system (Agilent Technologies, Santa Clara, CA, USA). The index codes were added to attribute sequences to each sample, and the cDNA libraries constructed by the above reaction system were performed on an Illumina HiSeq X Ten platform by Frasergen Bioinformatics Co., Ltd. (Wuhan, China). As a result, 150-bp paired-end reads were generated and then entered the filtering quality control process.

The clean reads were aligned to the reference genome *Chinese mitten crab* (*Eriocheir sinensis*) (https://www.ncbi.nlm.nih.gov/assembly/GCF_024679095.1) using HISAT 2 software [22]. Use Samtools to compress and sort mapping files, and use String Tie software for read assembly and abundance estimation [23]. In this experiment, the fragments per kilobase of exon per million fragments mapped reads (FPKM) represented the relative expression of genes, and FDR was a prerequisite for determining whether it was a differentially expressed gene (DEG). The list of DESeq2 identified DEGs, and the screening criteria for DEGs was FDR < 0.05 and fold change ≥ 2. GO (Gene Ontology) and KEGG (Kyoto Encyclopedia of Genes and Genomes) pathway enrichment was performed using phyper R package.

## Metabolic profiling

100 mg powdered samples were homogenized and were extracted with 1.0 mL pre-chilled 80% methanol solution containing 0.1% (v/v) formic acid at 4˚C overnight. The extracts were centrifuged for 10 min at 4˚C, 10,000 g, then absorbed and diluted into a concentration containing 53% methanol. The supernatant was filtrated before LC-ESI-MS/MS system (HPLC, Shim-pack UFLC SHIMADZU CBM30A system; MS, Applied Biosystems 4500 Q TRAP). LC–MS/MS analyses were performed using an ExionLCTM AD system coupled with a QTRAP® 6500 + mass spectrometer (both from SCIEX, Framingham, MA, USA). It described the system including the high performance liquid chromatography (HPLC) and MS conditions by He et al [24].

Identification and quantification of metabolites were analyzed using the MultiaQuant and Analyst software (1.6.1). The multivariate analysis of metabolites, including model evaluation,

was conducted by the orthogonal partial least squares discriminant analysis (OPLS-DA) with score plots and permutation plots. The differential metabolites were screened by combining the differential multiple, P value of the t-test and VIP value of the OPLS-DA model, and the screening standard was FC > 1, P value < 0.05 and VIP > 1. The identified metabolites were aligned to GO and KEGG database, and the significantly enriched functions explained the differences between sample groups.

## Results

### Transcriptome (RNA-seq) analysis

**Data acquisition of transcriptome.** In this study, transcriptome data of hepatopancreas (L) and ovary (R) were statistically analyzed. A total of 18 experimental samples generated over 148.12 Gb of clean data. The clean data from each sample reached at least 6.08 Gb and the percentage of Q30 bases was 93.90% and above (Table 1). Mapping ratio between clean reads and the reference genome of *Eriocheir sinensis* (https://www.ncbi.nlm.nih.gov/assembly/GCF_024679095.1) ranged from 80.27 to 88.34%, with an average of 84.86% (Table 1). Principal component analysis (PCA) showed significant differences between the the untreated group (C) and the treat groups (30 m, 60 m) in hepatopancreas and ovary, respectively. The changes after treatment were not uniform, and the repeatability within each group was acceptable. PC1 (40.44%) and PC2 (25.09%) effectively separated the control samples from the treatment samples in hepatopancreas (Fig 1A), and PC1(39.64%) and PC2 (24.14%) separated the control samples from the treatment samples in ovary (Fig 1A). The above results demonstrated that the processing method and the data obtained could be used.

**Analysis of differentially expressed genes.** Differentially expressed genes (DEGs) defined by fold change (FC) $\geq$ 2 and FDR < 0.01 reflected the gene expression patterns between different sample groups, therefore, providing some explanations for the body's response within 1 hour of AFB1 stress. In hepatopancreas and ovaries, AFB1 for 30 minutes and 45 minutes

**Table 1. Data statistics of transcriptome sequencing.**

| Sample ID | Clean Reads Pairs | Clean base(bp) | Q30(%) | Total mapped ratio |
|---|---|---|---|---|
| L-C-1 | 20,293,546 | 6,088,063,800 | 94.10 | 88.18% |
| L-C-2 | 20,985,943 | 6,295,782,900 | 93.90 | 88.13% |
| L-C-3 | 23,912,399 | 7,173,719,700 | 94.50 | 84.87% |
| L-30m-1 | 32,656,966 | 9,797,089,800 | 95.10 | 86.89% |
| L-30m-2 | 23,181,660 | 6,954,498,000 | 94.40 | 86.48% |
| L-30m-3 | 24,975,913 | 7,492,773,900 | 95.50 | 88.34% |
| L-60m-1 | 27,668,533 | 8,300,559,900 | 95.10 | 81.33% |
| L-60m-2 | 29,805,125 | 8,941,537,500 | 95.00 | 83.18% |
| L-60m-3 | 25,126,846 | 7,538,053,800 | 95.00 | 80.27% |
| R-C-1 | 29,744,172 | 8,923,251,600 | 94.50 | 83.81% |
| R-C-2 | 23,816,503 | 7,144,950,900 | 94.50 | 84.79% |
| R-C-3 | 32,709,317 | 9,812,795,100 | 94.40 | 85.04% |
| R-30m-1 | 28,947,683 | 8,684,304,900 | 94.10 | 87.06% |
| R-30m-2 | 27,988,337 | 8,396,501,100 | 94.20 | 82.97% |
| R-30m-3 | 32,120,101 | 9,636,030,300 | 94.30 | 81.86% |
| R-60m-1 | 32,180,957 | 9,654,287,100 | 94.80 | 86.13% |
| R-60m-2 | 31,520,243 | 9,456,072,900 | 94.20 | 81.40% |
| R-60m-3 | 26,093,048 | 7,827,914,400 | 94.0 0 | 86.81% |

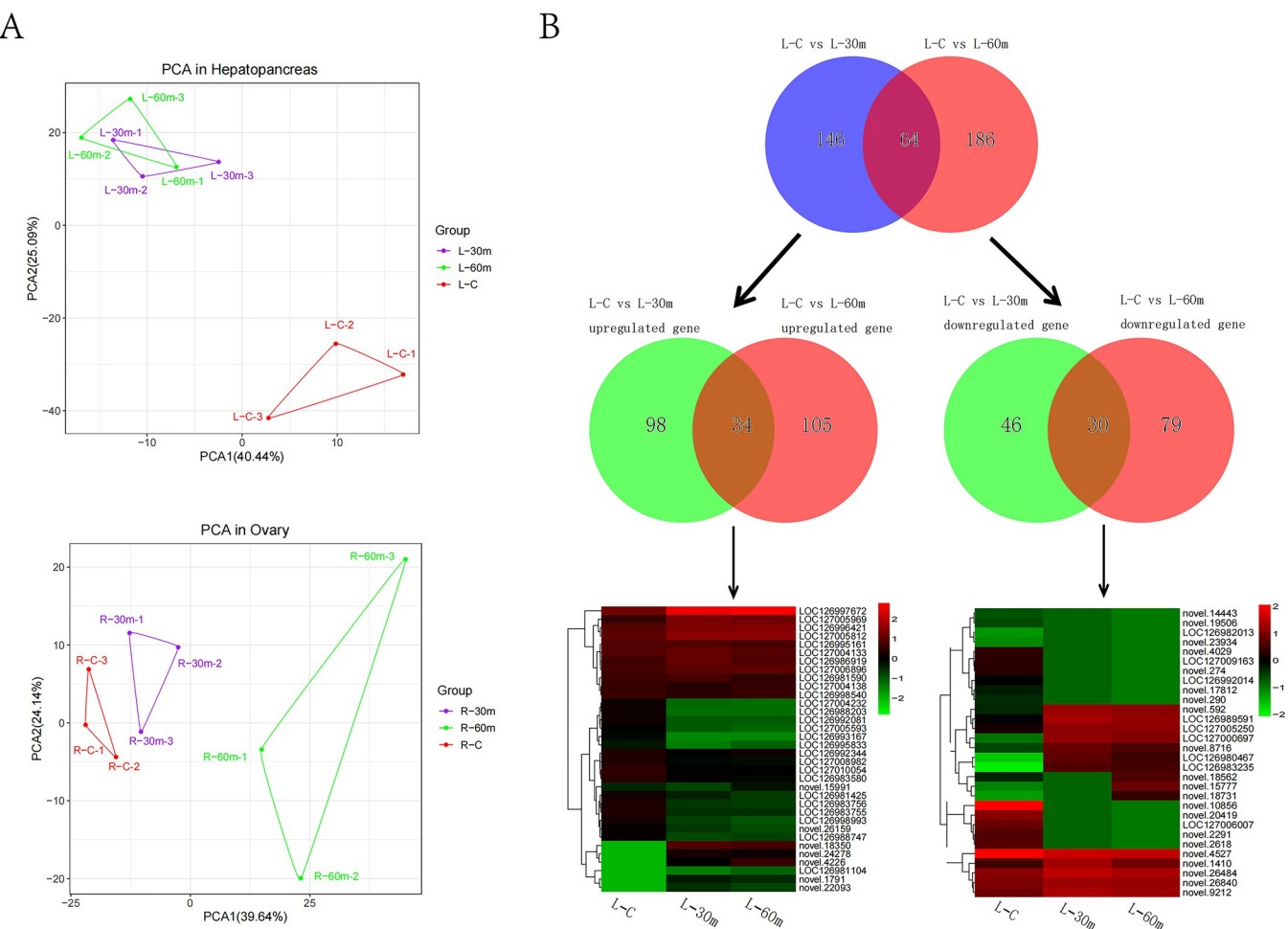

**Fig 1. PCA in hepatopancreas and ovary and DEGs in hepatopancreas. (A)** PCA showed differences between the untreated groups (C) and the treated groups (30 m, 60 m) in hepatopancreas and ovary. **(B)** DEGs generated between C vs. 30 min and C vs. 60 min in hepatopancreas, as well as venns and heatmaps generated between C vs. 30 min and C vs. 60 min.

groups were compared with control group, respectively. Specifically, we performed the conventional analysis of the pairwise comparisons of C vs. 30 min and C vs. 60 min in hepatopancreas and ovaries, respectively. Results showed, 210 (133 up-regulated; 77 down-regulated) and 250 (140 up-regulated; 110 down-regulated) DEGs were identified in the L-C vs. L-30 m and L-C vs. L-60 m comparison in hepatopancreas, respectively. Further analysis revealed 64 DEGs were the same in L-C vs. L-30 m and L-C vs. L-60 m comparison, with 34 DEGs up-regulated and 30 DEGs down-regulated (Fig 1B, S1 and S2 Tables). The 34 common up-regulated DEGs mentioned above had attracted our attention, guiding us to understand which genes were more active and which functions were enhanced after stress (Fig 1B). Studies had found that 14 up-regulated DEGs showed relatively complete functional annotation information or biological function related to this study (S3 Table). Based on the relevant gene annotations and functional gene research reports in the literature, we had simply classified the above 14 DEGs into six categories of functions, including antibacterial and detoxification, ATP energy reaction, redox reaction, nerve reaction, liver injury repair, and immune reaction. Among them, five DEGs were related to antibacterial and detoxification, followed by two DEGs related to ATP energy reaction, redox reaction, nerve reaction, and liver injury repair, and one DEG

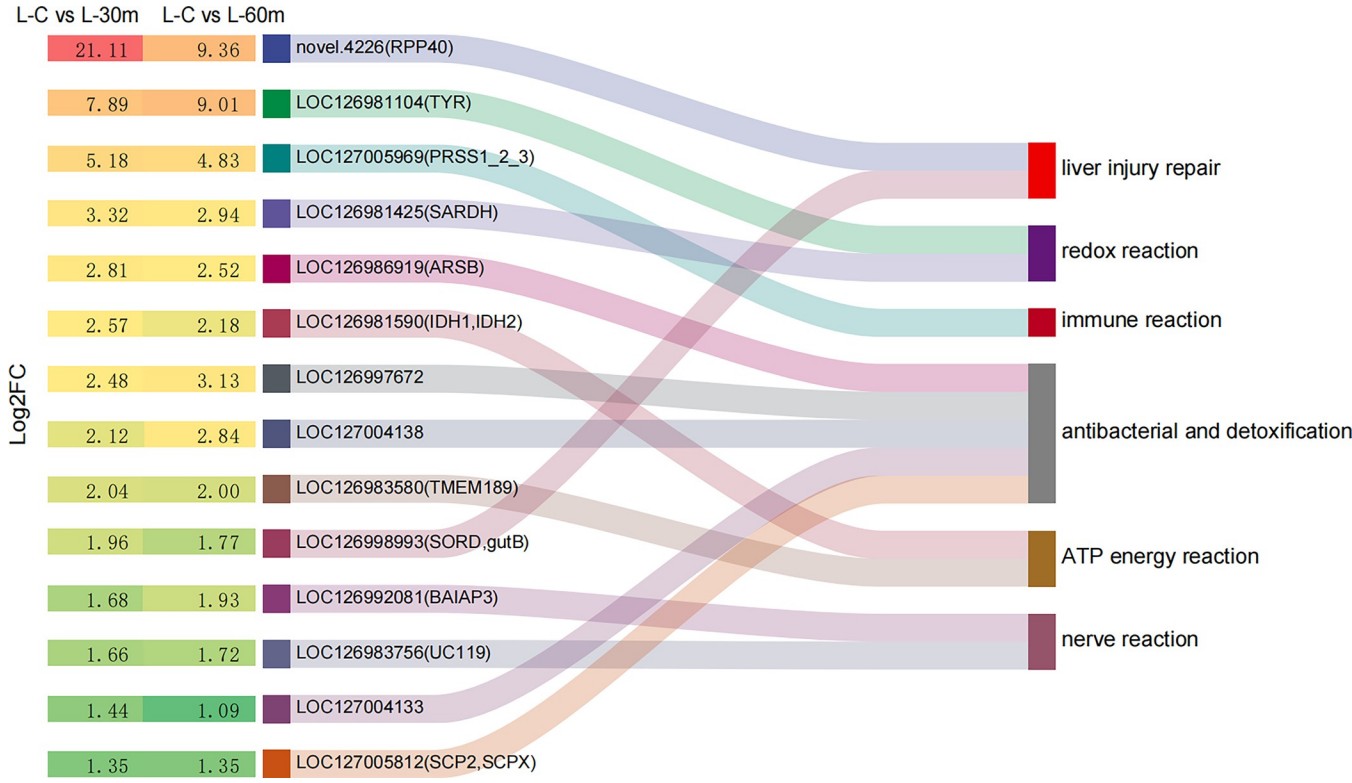

**Fig 2. Six categories of functions were classified by 14 DEGs, with Log2FC in C vs. 30 min and C vs. 60 min.**

related to immune reaction (Fig 2). The log2FC values of 14 DEGs ranged from 1.35 to 21.11, and the log2FC values of same gene between L-C vs. L-30 m and L-C vs. L-60 m was not significantly different, indicating that the possibility of individual differences leading to differential genes was very small (Fig 2). Literature showed that gene *LOC127005969 (PRSS1_2_3)* encoded trypsin-like serine proteinase and played an important role in the repair of pathological processes such as coagulation, immune response, fibrinolysis, inflammation and tumor [25]. The log2FC values of gene *LOC127005969* were 5.18 (L-C vs. L-30 m) and 4.83 (L-C vs. L-60 m), showing a significant increase in the treated group compared with the untreated group. Gene *LOC126998993* (*SORD*) encoded sorbitol dehydrogenase, and high expression of this protein indicated a signal of liver injury [26]. The expression level of *LOC126998993* increased nearly fourfold before and after treatment (the log2FC values of 1.96 in L-C vs. L-30 m and 1.77 in L-C vs. L-60 m, respectively). Gene *LOC127004138* could encode anti-lipopolysaccharide factor, which was widely used in aquaculture due to its strong antibacterial activity and broad antibacterial spectrum [27]. The expression level of *LOC127004138* increased more than fourfold before and after treatment (the log2FC values of 2.12 in L-C vs. L-30 m and 2.84 in L-C vs. L-60 m, respectively). Additionally, the increased expression of several proteases or enzymes reflects the enhanced detoxification ability of hepatopancreas to cope with the invasion of AFB1, such as lipid-transfer protein encoded by gene *LOC127005812* (*SCP2*); arylsulfatase encoded by gene *LOC126986919* (*ARSB*); hormone esterase encoded by gene *LOC127004133* and copper-specific metallothionein encoded by gene *LOC126997672*. All these proteins had been reported to be associated with detoxification [28], and in this study, the expression levels of these genes in the treated group increased by at least two times compared to the untreated group. In addition to these genes mentioned above, we also studied and

analyzed other biological activities, such as redox reactions, neural reactions, and energy activities.

In the ovaries, 124 (45 up-regulated; 79 down-regulated) and 99 (68 up-regulated; 31 down-regulated) DEGs were identified in the comparison of R-C vs. R-30 m and R-C vs. R-60 m, respectively (Fig 3A). The 124 and 99 DEGs mentioned above have a total of 12 shared DEGs, of which 7 DEGs were up-regulated and 5 DEGs were down-regulated (Fig 3B, S4 Table). NR annotation indicated most of DEGs were associated with proteins that have not been characterized or hypothesized. Only four genes (*LOC127003043*; *novel.27046*; *novel.898*; *LOC127005871*) were annotated in GO terminology, and GO explained the function of these genes as integral component of membrane. In the experiment on the ovaries, there were no characteristic DEGs presented, which reflected that within 1 hour of AFB1 invasion, the stress response activity of the ovaries was much less intense than that of the hepatopancreas.

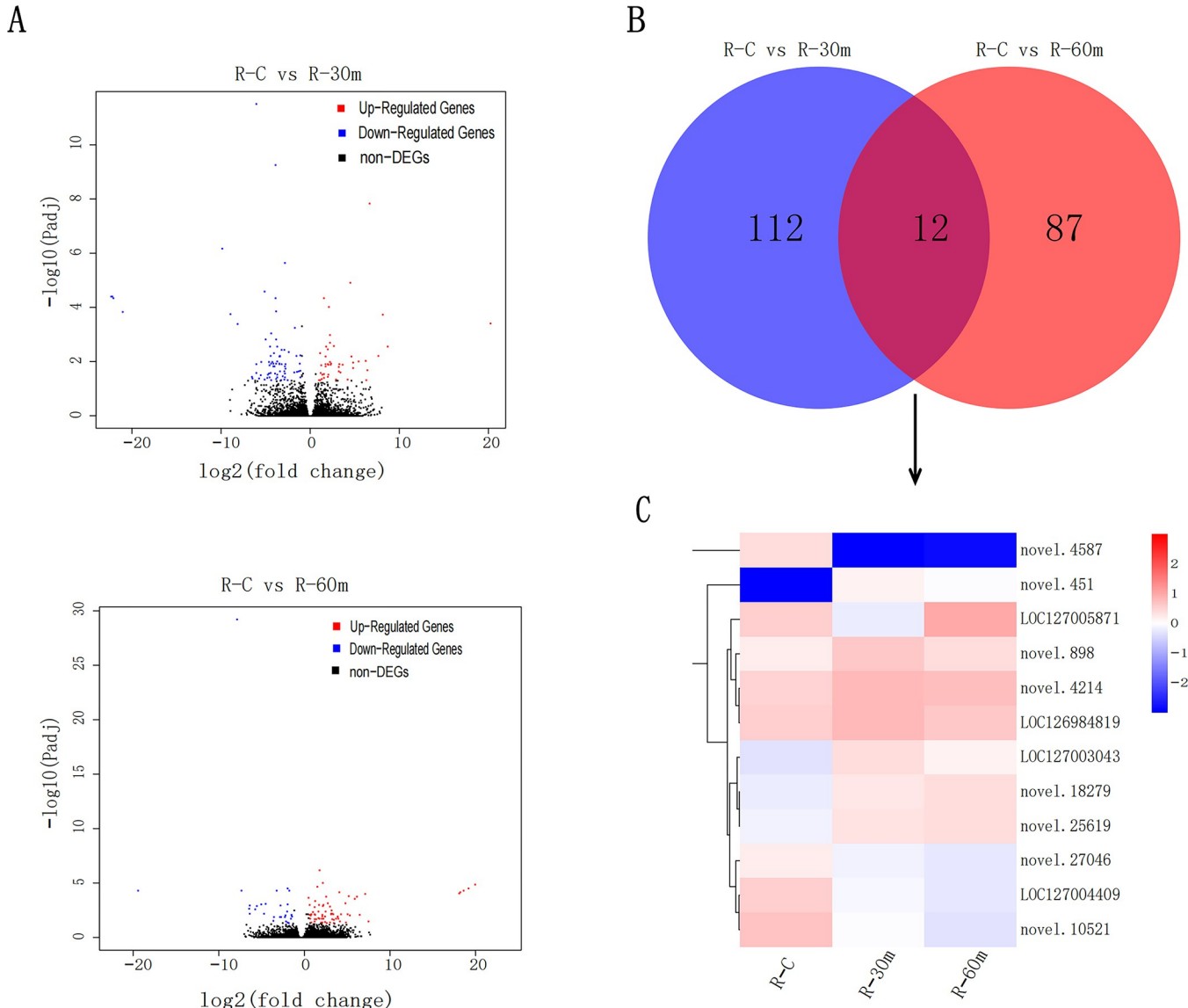

**Fig 3. DEGs generated in ovary. (A)** Volcanic maps of DEGs generated by C vs. 30 min and C vs. 60 min in the ovaries. **(B)** venns and heatmaps generated between C vs. 30 min and C vs. 60 min in ovary.

**K-means analysis in hepatopancreas.** To decipher the overall trend in gene expression profiles, k-means clustering method was applied to determine the trend change of DEGs in hepatopancreas. We subjected 318 DEGs in hepatopancreas to k-means clustering process and obtained five distinct subclusters (Fig 4). As a result, subcluster 2 revealed that the expression levels of 49 DEGs decreased after 30 minutes and remained basically unchanged from 30 minutes to 60 minutes. Subcluster 3 revealed the expression levels of 61 DEGs continued to increase from the beginning to 60 minutes. The expression levels of 44 DEGs showed a trend of first decreasing and then increasing from 30 minutes to 60 minutes (subcluster 1), while 23 DEGs first increased and then decreased from 30 minutes to 60 minutes (subcluster 5).

## Metabolomics analysis

**Analysis of metabolites based on UPLC-MS/MS-based quantitative.** To further explore the reactions in hepatopancreas and ovary within 1 hour after injection of AFB1 standard solution, an untargeted metabolomics method were used to detect metabolite compounds and investigate differential metabolites produced in the pairwise comparisons. In this study, a total of 13,963 metabolites were detected or identified, of which 6,642 were positive model and 7,321 were negative model. The number of metabolites for secondary identification was 4,829 in positive model and 4,003 in negative model, so we selected metabolites from positive model for subsequent analysis. Statistical analysis revealed that the components of metabolites were classified into 25 major categories, with amino acid and its metabolites accounting for 27.33%, followed by benzene and substituted derivatives (12.41%), and heterocyclic compounds (11.4%)(Fig 5A). The differential metabolites were searched by the model for orthogonal partial least squares discriminant analysis (OPLSDA), which performed 200 random permutations and combination experiments on the data. The OPLSDA results show that the model parameters for R2Y and Q2 values were close to 1, further verifying the reliability of the OPLSDA Model (S1 and S2 Figs).

**Functional annotation and enrichment analysis of differential metabolites.** Combined with the multivariate analysis of OPLS-DA, the variable importance projection (VIP) > 1 together with fold change $\geq 2$ or $\leq 0.5$, was used as the screening criteria for differential abundant metabolites. In a hepatopancreatic positive model, group using the blank working solution (ML-C) was compared with groups using the standard AFB1 working solution (ML-30 m and ML-60 m), resulting the differentially accumulated metabolites (DAMs). The OPLS-DA model maximized the discrimination between group using the blank working solution (ML-C) and groups using the standard AFB1 working solution (ML-30 m and ML-60 m), showing significant differences (Fig 5B). In the comparison between ML-C and ML-30 m, a total of 228 metabolites (133 up-regulated and 95 down-regulated) showed significant differences. A total of 401 metabolites (339 up-regulated and 62 down-regulated) were identified as DAMs in the ML-C vs. ML-60 m comparison (Fig 5C). According to the functional annotation of DAM, based on the Kyoto Encyclopedia of Genes and Genomes (KEGG) database, 25 and 31 pathways were enriched in the comparison of ML-C vs. ML-30 m and ML-C vs. ML-60 m, respectively (S5 and S6 Tables). KEGG pathway analysis founded they both enriched a total of 12 pathways, with clear functions of cutin, suberine and wax biosynthesis; tyrosine metabolism; purine metabolism; nucleotide metabolism; glycine, serine and threonine metabolism; ABC transporters and tryptophan metabolism. The significant enrichment pathways ($P<0.05$) in comparison between ML-C and ML-30 m were cyanoamino acid metabolism (ko00460), glutathione metabolism(ko00480) and sulfur relay system(ko04122) (Fig 6A). More specifically, N-hydroxy-L-isoleucine (MW0154477) and N-hydroxy-L-valine (MW0154482), as members of amino acid and its metabolites, were DAMs in the ML-C vs. ML-30 m comparison

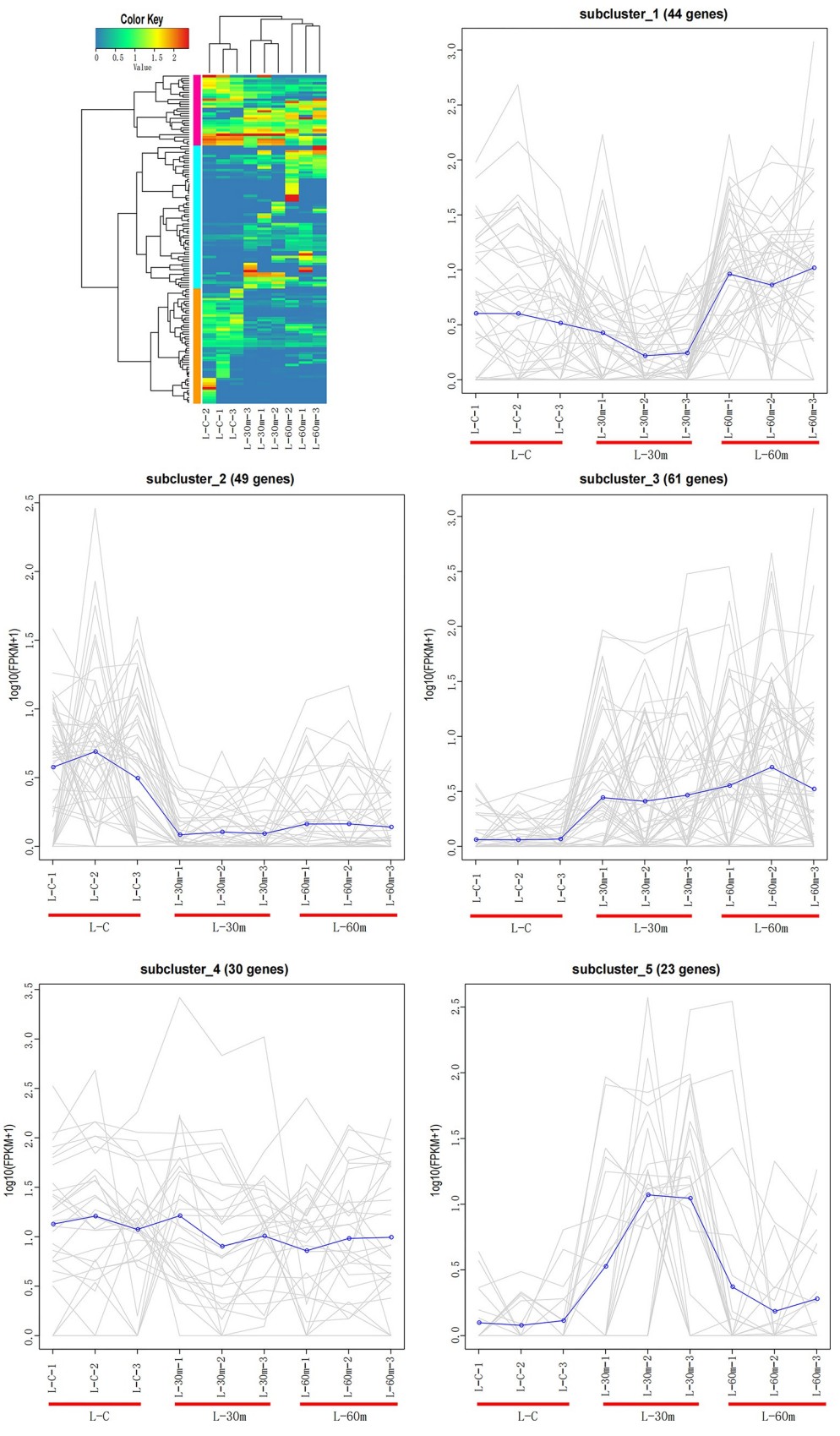

**Fig 4. K-means analysis of hepatopancreas.**

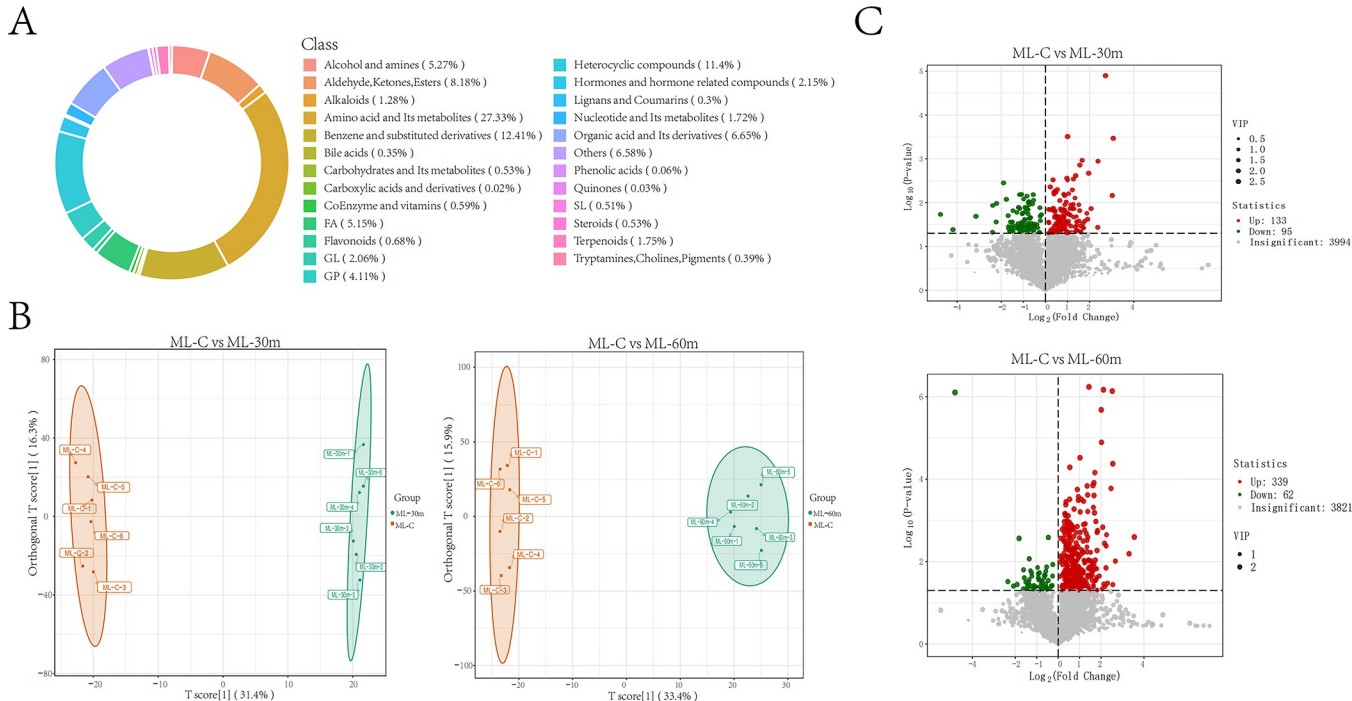

**Fig 5. Characteristics of metabolites and DAMs in hepatopancreas. (A)** 25 major categories were classified by metabolites. **(B)** The OPLS-DA model showed significant differences between the untreated groups (C) and the treated groups (30 m, 60 m) in hepatopancreas. **(C)** Volcanic maps of DAMs generated by C vs. 30 min and C vs. 60 min in hepatopancreas.

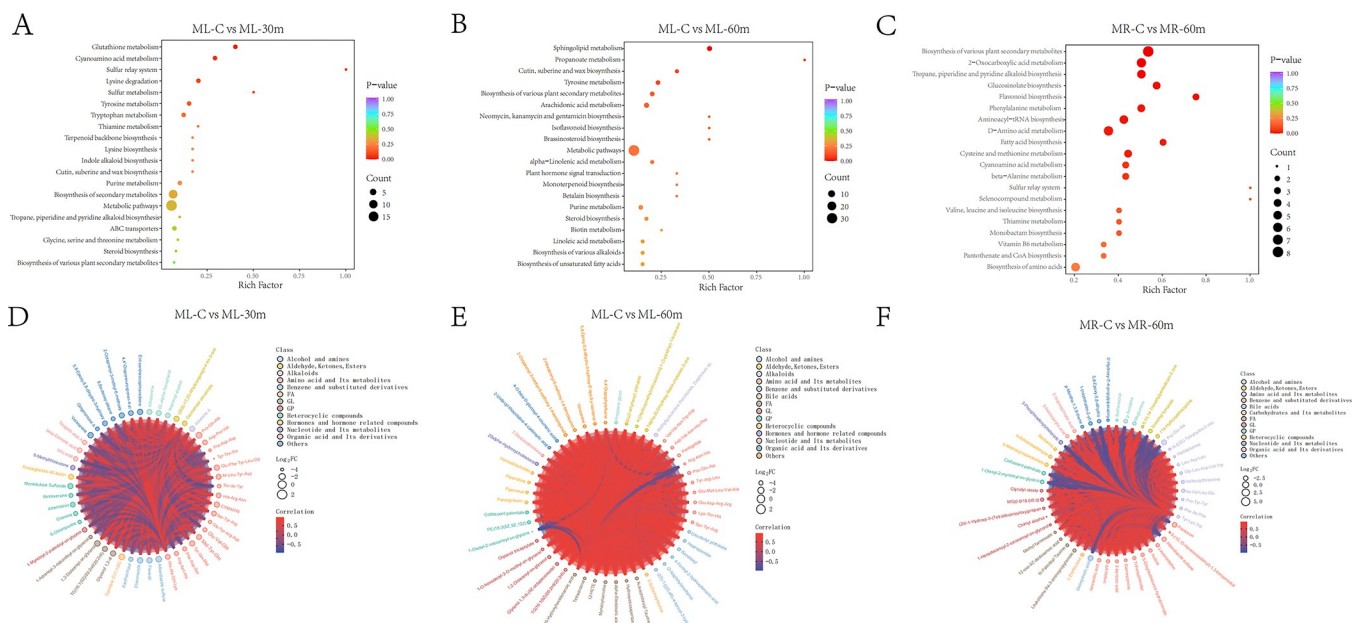

**Fig 6. Enrichment analysis of KEGG pathway and pearson correlation analysis of the top 50 differential metabolites in hepatopancreas and ovaries. (A)** Enrichment analysis of KEGG pathway for C vs. 30 min in hepatopancreas. **(B)** Enrichment analysis of KEGG pathway for C vs. 60 min in hepatopancreas. **(C)** Enrichment analysis of KEGG pathway for C vs. 60 min in ovary. **(D)** Pearson correlation analysis of the top 50 differential metabolites for C vs. 30 min in hepatopancreas. **(E)** Pearson correlation analysis of the top 50 differential metabolites for C vs. 60 min in hepatopancreas. **(F)** Pearson correlation analysis of the top 50 differential metabolites for C vs. 60 min in ovary.

and participated in the ko00460 pathway. In addition, gamma-glutamylalanine (MW0106760) and cadaverine (MW0110603) participated in the ko00480 pathway, and thiamine (MEDP0514) participated in the ko04122 pathway. In order to understand the synergistic or mutually exclusive relationship between different metabolites, the interactions between the top 50 differential metabolites were subjected to process by the Pearson correlation analysis method. The results showed that amino acid and its metabolites accounted for the highest proportion, with a total of 17 DAM members (Fig 6D).

The significantly enrichment pathways (P<0.05) in ML-C vs. ML-60 m comparison were sphingolipid metabolism (ko00600) (Fig 6B). Sphinganine(MW0111287),N-(dodecanoyl)-sphing-4-enine-1-phosphocholine (MW0055323) and phytosphingosine (MEDP1685), as members of sphingolipids (SL), were DAMs in ML-C vs. ML-60 m comparison and participated in the ko00600 pathway. The top 50 differential metabolites contained ten kinds of amino acid and its metabolites, seven kinds of fatty acyls (FA), and six kinds of benzene and substituted derivatives (Fig 6E).

In the ovarian positive model, group using the blank working solution (MR-C) in the ovaries was compared with groups using the standard AFB1 working solution (MR-30 m and MR-60 m), generating 262 (145 up-regulated and 117 down-regulated) and 452 (91 up-regulated and 361 down-regulated) differential metabolites in the comparison of MR-C vs. MR-30 m and MR-C vs. MR-60 m, respectively. In the comparison between MR-C and MR-30 m, no significant enrichment pathways were found in the 36 enriched KEGG pathways (P<0.05), while in the comparison between MR-C and MR-60 m, 10 significantly enriched KEGG pathways (P<0.05) were found in the 66 enriched KEGG pathways, including 2-oxocarboxylic acid metabolism, tropane, piperidine and pyridine alkaloid biosynthesis, glucosinolate biosynthesis, flavonoid biosynthesis, phenylalanine metabolism, aminoacyl-tRNA biosynthesis, D-amino acid metabolism, fatty acid biosynthesis, and cysteine and methionine metabolism (Fig 6C). The top 50 differential metabolites contained 11 kinds of benzene and substituted derivatives, followed by 10 kinds of amino acid and its metabolites (Fig 6F).

**Integrated analysis of metabolomics and transcriptomics.** The comprehensive analysis of metabolomics and transcriptome could systematically study the coordinated response of crabs to AFB1 stress. The combined data of multiple genomics could further analyze the interaction between genes and metabolites, so as to analyze and deal with this stress response. According to the data of metabolomics and transcriptomics, the DEGs and DAMs derived from the comparisons of ML-C vs. ML-30 m and ML-C vs. ML-60 m were Co-enriched and analyzed. The results showed that in ML-C and ML-30 m, 11 DEGs and 12 DAMs, were Co-enriched into 7 pathways in KEGG enrichment pathways (S7 Table); in the comparison between ML-C and ML-60 m, 17 DEGs and 19 DAMs Co-enriched into 9 pathways (S8 Table). Among the 7 and 9 pathways mentioned above, there are four pathways, including glycine, serine and threonine metabolism, tryptophan metabolism, tyrosine metabolism and purine metabolism, which were shared by both (Fig 7A). In order to further explore the role of genes and metabolites in the same pathway, we anchored the up-regulated DEGs and DAMs at specific locations and studied their interactions. From certain positions in the three pathways of steroid biosynthesis (ko00100), glycine, serine and threonine metabolism (ko00260), and sphingolipid metabolism (ko00600), there were up-regulated DEG and DAM involved in co- regulation, which had caught our attention. In steroid biosynthesis (ko00100) pathway, enzyme sterols were products of transcription-translation and modification by gene *LOC127008729* (*EGR3*), which catalyzed the conversion of episterol into ergostathenol (Fig 7B). Ergostathenol was converted to ergostatetraenol (C05440) under a action of the isoenzymes of EGR, and then into ergosterol. Ergosterol, as an important source of fat-soluble vitamin D2, had significant antibacterial and anti-tumor effects [29, 30]. In this pathway, the

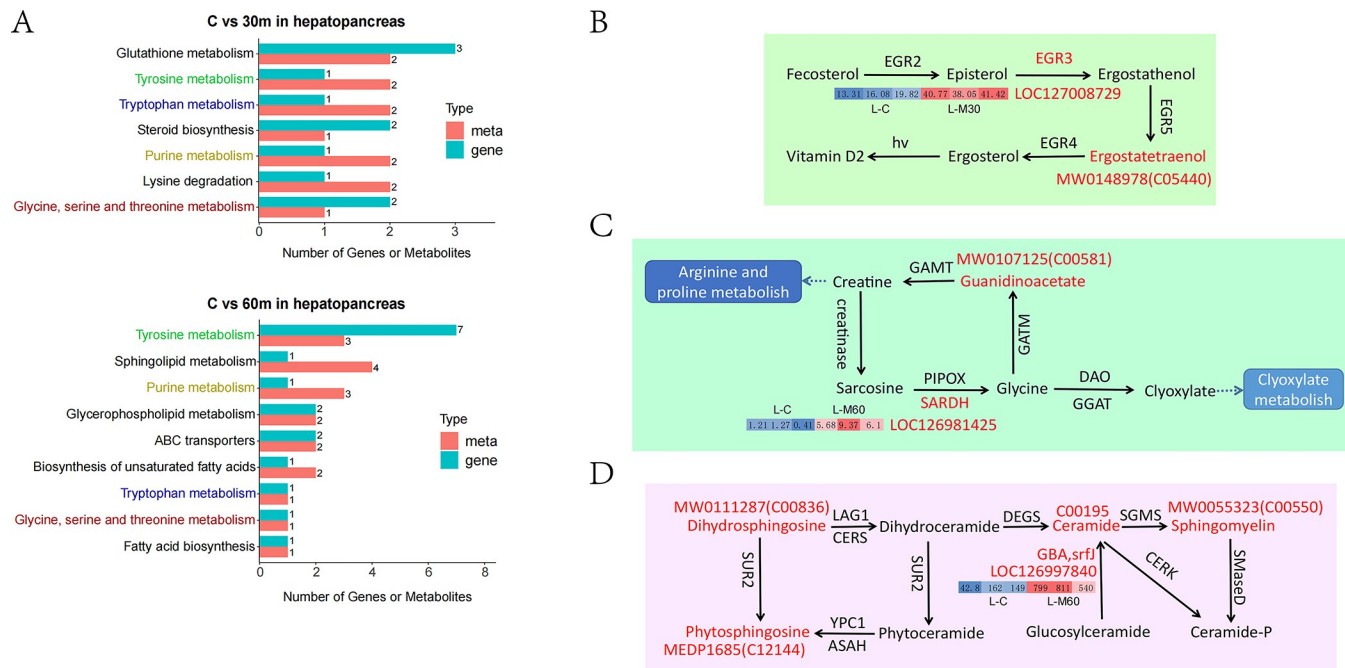

**Fig 7. KEGG pathways Co-enriched by DEGs and DAMs and interaction signals generated by up-regulated DEGs and DAMs. (A)** KEGG pathways Co-enriched by DEGs and DAMs between the comparison of C vs. 30 min and C vs. 60 min in hepatopancreas. **(B)** Interaction signals at specific local locations in steroid biosynthesis pathway. **(C)** The interaction signals at specific local locations in glycine, serine and threonine metabolism. **(D)** The interaction signals at specific local locations in sphingolipid metabolism.

mean FPKM of DEG EGR3 in the L-C group and L-30m group were 16.40 and 40.08, respectively. The expression level of DAM MW0148978 (C05440) significantly increased from ML-C stage to ML-30 m stage and showed differences. The second ko00260 pathway showed that DEG *LOC126981425* (*SARDH*) encoded sarcosine dehydrogenase, which catalyzed of sarcosine to glycine (Fig 7C). Recent studies have found that the mechanisms of glycine included physiological protection against ischemia reperfusion injury, shock, transplantation, alcoholic hepatitis, liver fibrosis, arthritis, tumor, and drug toxicity, as well as anti-inflammatory and immunomodulatory effects [31, 32]. A portion of glycine was involved in clyoxylate metabolism, and the other portion was converted to guanidinoacetate (C00581) under the action of GATM, ultimately converting to creatine and entering arginine and proline metabolism. Creatine contributed to the circulation of adenosine triphosphate (ATP), thereby facilitating energy supply to muscles and the brain. Arginine could promote cell proliferation, improve animal growth performance, enhance immunity and antioxidant capacity [33, 34]. The mean FPKM of DEG SARDH in the L-C group and L-60m group were 0.96 and 7.05, respectively, indicating that the expression response of this gene after stress was quite intense. Meanwhile, the expression level of DAM MW0107125 (C00581) also sharply increased in the ML-60 m group compared to ML-C group. The third ko00600 pathway showed that DEG *LOC126997840* (*GBA,srfJ*) was a highly expressed gene in the untreated group (L-C), with an average FPKM of 117.93. However, in the 60 minute treatment group (L-60 m), the mean FPKM was as high as 716.67, indicating a significant change in the expression level of this gene after stress. The expression levels of DAM C00195, MW0111287 (C00836), MEDP1685 (C12144) and MW0055323 (C00550) co-expressed in this pathway also significantly increased after 60 minutes of treatment (Fig 7D). Phytosphingosine (C12144), as the precursor of ceramide (C00195), was a natural broad-spectrum antibacterial agent. Sphingomyelin (C00550)

regulated immune function and inflammatory reaction, and participated in oxidative stress reaction [35]. Many cellular stress inducers, such as inflammatory activation, excessive saturated fatty acid intake, and chemotherapy, could lead to an increase in the rate of ceramide (C00195) synthesis [36, 37].

## Discussion

In this study, we integrated transcriptomics and metabolomics analysis of hepatopancreas and ovary, and looked for the regulatory mechanism of *Eriocheir sinensis*'s adaptability to aflatoxin B1 from the perspective of differential expression of genes and metabolites. DEGs and DAMs revealed the differential response between groups without AFB1 and groups with AFB1, which helps us analyze and manage this stress response. Transcriptomics results showed that a series of complex changes occurred in hepatopancreas and ovary within one hour after AFB1 injection. The hepatopancreas, as an immune and detoxifying organ, has a natural response to AFB1, as evidenced by changes in the expression of many functional genes. However, we did not find this strong response in the ovaries. We speculate that within one hour of being invaded by AFB1 in crabs, the hepatopancreas have already participated in various regulatory responses to cope with this stress, while the ovaries are not the main response organs during this period, and their responses are not very sensitive.

Compare the blank working solution groups (L-C) and standard AFB1 working solution groups (L-30 m and L-60 m) in hepatopancreas, and annotate the generated DEGs into GO, KEGG, and NR databases, respectively, to screen the target genes as the focus of this study. The hepatopancreas is the main detoxifying organ and the main site for material metabolism and transformation, making it highly susceptible to damage from toxic substances [3]. According to reports, AFB1 induced hepatopancreatic injury is mainly mediated by four mechanisms: CYP450 enzyme, mitochondria, immune system, and free radicals [38]. In our sequencing results, we found that the gene novel.6517 and the gene *LOC126984017* (*UGT*) were associated with the synthesis of eytoehrome P450. Gene *novel.6517* encoded NADPH—cytochrome P450 reductase, with an mean FPKM of 0.01 and 2.50 in the L-C and L-60 m groups, respectively. The mean FPKM of gene *LOC126984017* (*UGT*) in L-C group and L-30m group were 18.30 and 65.73, respectively, and it is involved in the metabolism of xenobiotics and drug by cytochrome P450. The abnormal increase in the expression levels of these two genes indirectly indicates that P450 enzymes are more active in response during stress.

Further systematic analysis showed there were 64 identical DEGs in the comparison between L-C vs. L-30 m and L-C vs. L-60 m, of which 34 identical DEGs were up-regulated. Based on functional annotation information and literature reports, we dissected 14 DEGs and classified them into six categories of functions, including antibacterial and detoxification, ATP energy reaction, redox reaction, nerve reaction, liver injury repair and immune reaction. According to reports, AFB1 induced immune toxicity reactions may lead to dysfunction of oxidative stress function [39]. During this process, we discovered two DEGs (*LOC126981104*, *LOC126981425*) associated with oxidative reactions and one DEG (*LOC127005969*) associated with immune responses, indicating that AFB1 can cause an increase in the expression of redox related genes, thus supporting the viewpoint reported in the above literature. In practical applications, antioxidants can be added to animal feed to protect and prevent AFB1 poisoning in livestock and poultry by enhancing antioxidant capacity and immune function [40]. The literature indicates that an increase in the expression levels of two DEGs (*LOC126998993*, *novel.4226*) indicates liver injury, indicating that these two genes have an self-evident repair effect on liver tissue [26]. The abnormal expression of four hydrolytic enzymes related to regulatory effects encoded by DEGs (*LOC127004133*, *LOC127005812*, *LOC126997672*,

*LOC127004138*) is related to the antibacterial and detoxification of the liver, and has also been mentioned in the literature [27, 28]. Additionally, DEGs are also involved in neuromodulation and energy response, which are also synergistic reactions of hepatopancreas after stress.

The results of the metabolomics showed a total of 12 pathways were enriched in the comparison between ML-C vs. ML-30 m and ML-C vs. ML-60 m. Among them, cyanoamino acid metabolism (ko00460), glutathione metabolism(ko00480), sulfur relay system(ko04122) and sphingolipid metabolism (ko00600) were significantly enriched (P<0.05). Two DAMs (MW0106760, MW0110603) were involved in the glutathione metabolism, which has been reported to have antioxidant, immune, and detoxifying properties effects [41]. Three DAMs (MW0111287, MW0055323, MEDP1685) were involved in the sphingolipid metabolism, and sphingolipids have been reported to involved in many important signal transduction processes, including regulating cell growth, differentiation, aging, and programmed cell death [42]. From the above enrichment analysis results, it can be inferred that the expression of metabolites in hepatopancreas is also actively responding to AFB1 stress.

The Co-enriched pathways of metabolomics and transcriptomics were subjected to explore the interaction mechanism between them. The seven pathways Co-enriched by DEGs and DAMs between the untreated group (C) and the 30 min treat groups (30 m) and the nine pathways Co-enriched by DEGs and DAMs between the untreated group (C) and the 60 min treat groups (60 m) shared four identical pathways, including glycine, serine and threonine metabolism, tryptophan metabolism, tyrosine metabolism and purine metabolism. The metabolic enrichment of the above three amino acids reflects the synergistic response and consistent regulation of DEG and DAM when the hepatopancreas faces AFB1 stress. These feedbacks provide good research ideas and directions for future research on this stress regulation, especially in the discovery of important functions in the glycine, serine and threonine metabolism. DEGs and DAMs involved in this pathway are up-regulated, which will be further studied in detail in the following discussion. We anchored the up-regulated DEGs and DAMs on the KEGG map and depicted their interaction relationship on the map. Three interaction signals were generated from specific positions on the KEGG maps, including steroid biosynthesis (ko00100), glycine, serine and threonine metabolism (ko00260), and sphingolipid metabolism (ko00600).

In steroid biosynthesis, ergosterol is produced through the actions of DEG *LOC127008729* (*EGR3*) and DAM MW0148978 (C05440), which are reaction enzyme and substrate on the KEGG map, respectively. Ergosterol is an important component of fungal cell membranes and plays an important role in ensuring the integrity of membrane structure. Ergosterol can significantly inhibit tumor growth and activate inhibitory genes, therefore demonstrating a strong antibacterial ability. Research has shown that it can also enhance immune function and convert into fat soluble vitamin D2 [29, 30]. In the metabolism of glycine, serine, and threonine, DEG *LOC126981425* (*SARDH*) encodes sarcosine dehydrogenase, which is an important catalytic enzyme for the production of glycine. Glycine has anti-inflammatory and immunomodulatory effects, and literature has shown that it has physiological protective effects on liver injury and hepatitis [31, 32]. The metabolite guanidinoacetate represented by MW0107125 (C00581) on KEGG map will ultimately be converted into creatine, which then converts into arginine and proline metabolites. According to reports, arginine can improve animal growth performance, enhance immunity and antioxidant capacity. Therefore, it is often used as a feed additive, with effects such as wound healing, ammonia excretion, immune function, and hormone secretion [33, 34]. The synergistic up-regulation of *LOC126981425* (*SARDH*) and MW0107125 (C00581) in this pathway, combined with their functions, can infer the regulatory response and protective mechanism played by the hepatopancreas in crabs. In sphingolipid metabolism, the metabolites dihydrosphingosine, phytosphingosine, sphingomyelin and ceramide were all detected as DAMs on the ko00600 map. They can transform into each other

under certain conditions. phytosphingosine is produced as a natural broad-spectrum antibacterial agent; sphingomyelin and ceramide could regulate immune function, inflammatory reaction and oxidative stress reaction [35–37]. The function of these DAMs can also indicate an enhancement of hepatopancreatic sphingolipids metabolism to resist the invasion of AFB1.

This study explored genes or metabolites related to antibacterial activity, redox ability, liver injury protection, immune enhancement, and anti-inflammatory repair by combining target functional genes, metabolites, and metabolic pathways screened by multiple omics. These genes and metabolites play an important role in responding to the invasion of AFB1 in the hepatopancreas or ovaries of crabs. This experiment provides us with valuable insights into the molecular mechanism of the response of crab hepatopancreas to AFB1, and provides direction and ideas for future guidance in addressing AFB1 damage to crabs.

## Supporting information

**S1 Fig. The OPLS-DA model validation in ML-C vs. ML-30m comparison.**
(TIF)

**S2 Fig. The OPLS-DA model validation in ML-C vs. ML-60m comparison.**
(TIF)

**S1 Table. Up-regulated DEGs in comparison between L-C vs. L-30m and L-C vs. L-60m.**
(XLSX)

**S2 Table. Down-regulated DEGs in comparison between L-C vs. L-30m and L-C vs. L-60m.**
(XLSX)

**S3 Table. Cored and up-regulated DEGs during AFB1 injection in hepatopancreas.**
(DOCX)

**S4 Table. DEGs in comparison between R-C vs. R-30m and R-C vs. R-60m.**
(XLSX)

**S5 Table. KEGG pathways were enriched by DAMs in ML-C vs. ML-30m comparison.**
(XLSX)

**S6 Table. KEGG pathways were enriched by DAMs in ML-C vs. ML-60m comparison.**
(XLSX)

**S7 Table. KEGG pathways were Co-enriched by DEGs and DAMs in C vs. 30 m comparison.**
(DOCX)

**S8 Table. KEGG pathways were Co-enriched by DEGs and DAMs in C vs. 60 m comparison.**
(DOCX)

## Acknowledgments

The experimental design and data analysis of this article were guided by Professor Yifeng Li from College of Aquatic and Life Sciences, Shanghai Ocean University; Yitian technologiescorporation provided experimental equipment and materials for this experiment.

## Author Contributions

**Conceptualization:** Hongsheng Yang, Xiaohua Zhu.

**Data curation:** Hongsheng Yang, Shaofang He.

**Formal analysis:** Hongsheng Yang, Shaofang He.

**Funding acquisition:** Hongsheng Yang.

**Investigation:** Hongsheng Yang, Qiuyun Zhang, Xuguang Li, Shaofang He.

**Methodology:** Hongsheng Yang, Yifeng Li, Lei Wu, Shaofang He.

**Project administration:** Hongsheng Yang, Meifang Shen, Qiuyun Zhang, Yifeng Li, Lei Wu, Xiaohua Zhu.

**Resources:** Yifeng Li, Xuguang Li, Xiaohua Zhu.

**Software:** Xuguang Li, Huimin Chen, Lei Wu.

**Supervision:** Hongsheng Yang, Qiuyun Zhang, Xuguang Li, Xiaohua Zhu.

**Validation:** Hongsheng Yang, Lei Wu.

**Visualization:** Yifeng Li, Xiuhui Tan, Lei Wu, Xiaohua Zhu.

**Writing – original draft:** Hongsheng Yang, Xiuhui Tan.

**Writing – review & editing:** Hongsheng Yang.

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
