## [Decision Letter · Decision Letter 0]

5 Oct 2023

PONE-D-23-23404Transcriptome and metabolomics analysis of adaptive mechanism of Chinese mitten crab (Eriocheir sinensis) to aflatoxin B1PLOS ONE

Dear Dr. Zhu,

Thank you for submitting your manuscript to PLOS ONE. After careful consideration, we feel that it has merit but does not fully meet PLOS ONE’s publication criteria as it currently stands. Therefore, we invite you to submit a revised version of the manuscript that addresses the points raised during the review process.

We look forward to receiving your revised manuscript.

Kind regards,

Amel Mohamed El Asely

Academic Editor

PLOS ONE

Journal Requirements:

“This Study was supported in part by the 333 High-level Talent Training Project of Jiangsu Province (BRA2019093), the Jiangsu Independent Innovation Project of Agricultural Science and Technology (CX(19)3007), and the Jiangsu Modern Agricultural Industry Technology System Project (JATS[2022]371).”

5. We are unable to open your Supporting Information file [Supporting Information.rar and Figures.rar]. Please kindly revise as necessary and re-upload

6. Please include a copy of Tables 1-9 which you refer to in your text on page 27.

7. Please upload a copy of Figures 1-7, to which you refer in your text on page 28. If the figure is no longer to be included as part of the submission please remove all reference to it within the text.

Reviewers' comments:

Reviewer's Responses to Questions

**Comments to the Author**

1. Is the manuscript technically sound, and do the data support the conclusions?

Reviewer #1: Yes

Reviewer #2: Yes

2. Has the statistical analysis been performed appropriately and rigorously? 

Reviewer #1: N/A

Reviewer #2: Yes

3. Have the authors made all data underlying the findings in their manuscript fully available?

Reviewer #1: Yes

Reviewer #2: Yes

4. Is the manuscript presented in an intelligible fashion and written in standard English?

Reviewer #1: Yes

Reviewer #2: Yes

5. Review Comments to the Author

Reviewer #1: Grammatical and spelling mistakes

Alignment not proper

Scientific names should be in italics

Please change ul-µl throughout the manuscript

Comparison data, results, or references should be there in the Results section

Statistical data should be explained more properly

Line 48- Food and feed?

Line 58 and 59- Rewrite the sentence

Line 81- We hope? It is research work and if results are there then why expect?

Line 96 and Line 101- “certain amount”- Please mention the amount

Line 107-Line 109- After 30 minutes or 60 minutes? Give only one-time point

that was done for the experiment

Line111- “Frozen samples with liquid nitrogen”- Rewrite the sentence

Line 126- Different groups- Mention the groups

Line 139-Line 141-Rewrite the sentence-It is not clear

Line 149-Line 150- Rewrite the sentence-Not clear

Line 170-Line 173- Explain the statistical analysis in detail

Line 191- PC1 and PC2- What are these? Explain?

Line 201- Line 205- Break the sentences to make it more clear

Reviewer #2: The manuscript entitled “Transcriptome and metabolomics analysis of the adaptive mechanism of Chinese mitten crab (Eriocheir sinensis) to aflatoxin B1” is an interesting study and the authors have collected a unique dataset using a cutting-edge methodology. The paper is generally well-written and structured and is presented in an intelligible fashion written in Standard English. This is described in a technically sound piece of scientific research with data that supports the conclusions. Experiments were conducted rigorously, with appropriate controls, replication, and sample sizes. The conclusions are appropriately based on the data presented in an appropriate and rigorous statistical analysis.

6. PLOS authors have the option to publish the peer review history of their article (what does this mean?). If published, this will include your full peer review and any attached files.

Reviewer #1: No

Reviewer #2: No

---

## [Author Response · Author response to Decision Letter 0]

7 Nov 2023

To Journal Requirements:

We have made the necessary modifications according to the instructions.

We delete the ‘Funding Information’, and have received no specific funding for this work.

“This Study was supported in part by the 333 High-level Talent Training Project of Jiangsu Province (BRA2019093), the Jiangsu Independent Innovation Project of Agricultural Science and Technology (CX(19)3007), and the Jiangsu Modern Agricultural Industry Technology System Project (JATS[2022]371).”

We have modified the acknowledgment section and added this sentence to the cover letter,“The author(s) received no specific funding for this work.”

We create a ORCID iD for the corresponding author. ID number: 0000-0002-8595-5783.

5. We are unable to open your Supporting Information file [Supporting Information.rar and Figures.rar]. Please kindly revise as necessary and re-upload

We have uploaded each figure and table separately.

6. Please include a copy of Tables 1-9 which you refer to in your text on page 27.

Sorry, the Tables were in the compressed package, [Supporting Information rar.] We have re uploaded the tables.

7. Please upload a copy of Figures 1-7, to which you refer in your text on page 28. If the figure is no longer to be included as part of the submission please remove all reference to it within the text.

Sorry, the Figures were in the compressed package, [Figures.rar.] We have re uploaded the Figures.

8.Please review your reference list to ensure that it is complete and correct. If you have cited papers that have been retracted, please include the rationale for doing so in the manuscript text, or remove these references and replace them with relevant current references. Any changes to the reference list should be mentioned in the rebuttal letter that accompanies your revised manuscript. If you need to cite a retracted article, indicate the article’s retracted status in the References list and also include a citation and full reference for the retraction notice.

Not modified

To Reviewer #1

1.Grammatical and spelling mistakes

we check it and correct it

2.Scientific names should be in italics

species names, genes and other scientific names were italics in this revision.

Corrected.

3.Please change ul-µl throughout the manuscript

Corrected.

4.Comparison data, results, or references should be there in the Results section. Statistical data should be explained more properly

I'm sorry, I uploaded the figures and tables of the results to the website in a compressed package. However, for some reason, this file cannot be seen by you. This submission will upload each data result separately.

5.Line 48- Food and feed?

It mainly present in feed, delete “ food”.

6.Line 58 and 59- Rewrite the sentence 

The effects of AFB1 on aquatic animals have been widely reported in many research articles both domestically and internationally, such as Oreochromis niloticus [7,8], Oncorhynchus mykiss [9], Litopenaeus vannamei [10], Lctalurus punctatus [11], Labeo rohita [12] and other species.

7.Line 81- We hope? It is research work and if results are there then why expect?

This sentence is not appropriate. We have changed it to “ we analyzed the obtained data in this experiments and explained some molecular response mechanism in the results.

8.Line 96 and Line 101- “certain amount”- Please mention the amount

Line 96: by weighing 6 mg of aflatoxin B1 standard and dissolving it with 100 ml of dimethyl sulfoxide

Line 101 : “More than 18 female experimental crabs” 

9.Line 107-Line 109- After 30 minutes or 60 minutes? Give only one-time point that was done for the experiment

I rewrote this sentence. “The hepatopancreas and ovaries of experimental crabs injected with AFB1 in first group were taken out after 30 minutes, and those in the second group were taken out after 60 minutes”

10.Line111- “Frozen samples with liquid nitrogen”- Rewrite the sentence

The samples were frozen with liquid nitrogen and quickly stored in an ultra-low temperature refrigerator (-80℃)

11.Line 126- Different groups- Mention the groups

Correct into “the above three groups (one control and two treatments)”

12.Line 139-Line 141-Rewrite the sentence-It is not clear

The clean reads were aligned to the reference genome Chinese mitten crab (Eriocheir sinensis) (https://www.ncbi.nlm.nih.gov/assembly/GCF_024679095.1) using HISAT 2 software

13.Line 149-Line 150- Rewrite the sentence-Not clear

I rewrote this sentence: the screening criteria for DEGs was FDR < 0.05 and fold change ≥ 2.

14.Line 170-Line 173- Explain the statistical analysis in detail

I rewrote this sentence “The differential metabolites were screened by combining the differential multiple, P value of the t-test and VIP value of the OPLS-DA model, and the screening standard was FC > 1, P value < 0.05 and VIP > 1. ” T-test is the use of t-distribution theory to infer the probability of differences occurring. The screening of differential metabolites is usually based on the OPLS-DA model, and VIP is used to evaluate the importance of each metabolite in metabolomic data analysis of sample classification or prediction models, usually greater than 1.

15.Line 191- PC1 and PC2- What are these? Explain?

Principal Component Analysis (PCA) is the most widely used data dimensionality reduction algorithm. All features of the sample are reflected using two principal components, PC1 and PC2. The position relationship of samples on the coordinate axis reflects the similarity between samples. The closer the position is, the higher the similarity between the two samples. The numerical value represents the explanatory rate of variance, and the larger the numerical value, the more reliable the model is.

16.Line 201- Line 205- Break the sentences to make it more clear

I rewrote this sentence “In hepatopancreas and ovaries，AFB1 for 30 minutes and 45 minutes groups were compared with control group, respectively.”

---

## [Editor Report · Decision Letter 1]

20 Nov 2023

Transcriptome and metabolomics analysis of adaptive mechanism of Chinese mitten crab (Eriocheir sinensis) to aflatoxin B1

PONE-D-23-23404R1

Dear Dr. Xiaohua Zhu 

We’re pleased to inform you that your manuscript has been judged scientifically suitable for publication and will be formally accepted for publication once it meets all outstanding technical requirements.

Kind regards,

Amel Mohamed El Asely

Academic Editor

PLOS ONE

---

## [Editor Report · Acceptance letter]

28 Nov 2023

PONE-D-23-23404R1 

Transcriptome and metabolomics analysis of adaptive mechanism of *Chinese mitten crab* (*Eriocheir sinensis*) to aflatoxin B1 

Dear Dr. Zhu:

I'm pleased to inform you that your manuscript has been deemed suitable for publication in PLOS ONE. Congratulations! Your manuscript is now with our production department. 

Kind regards, 

on behalf of

Prof. Amel Mohamed El Asely 

Academic Editor

PLOS ONE